# Stress among Nursing Students in the Era of the COVID-19 Pandemic

**DOI:** 10.3390/healthcare12181885

**Published:** 2024-09-20

**Authors:** Grzegorz Kobelski, Katarzyna Naylor, Aleksandra Kobelska, Mariusz Wysokiński

**Affiliations:** 1Institute of Medical Sciences, The University College of Applied Sciences in Chelm, Pocztowa 54, 22-100 Chełm, Poland; 2Department of Didactics and Medical Simulation, Faculty of Medical Sciences, Medical University of Lublin Poland, Chodźki 7, 20-093 Lublin, Poland; katarzyna.naylor@umlub.pl; 3Municipal Independent Public Health Care Institution in Chelm, Wołyńska 11, 22-100 Chełm, Poland; olakobelska@gmail.com; 4Department of Fundamentals of Nursing, Faculty of Health Sciences, Medical University of Lublin Poland, Staszica 4/6, 20-093 Lublin, Poland; mariusz.wysokinski@umlub.pl

**Keywords:** stress, COVID-19 pandemic, students nursing

## Abstract

Introduction: The COVID-19 pandemic has significantly impacted stress levels across various professions, particularly in the medical field. This increase in stress has also affected medical students, including nursing students, who faced unprecedented and challenging circumstances. Nursing students, in particular, experienced added pressure due to observing the frontline experiences of nurses and the new demands placed upon them. Aim: This study aimed to assess whether the COVID-19 pandemic affected an increase in stress levels among nursing students in Poland during the pandemic. We also attempt to determine whether there is a correlation between the stress level of students and factors such as gender, age, place of residence, marital status, and level of education. Assuming that the stress level will be higher among women of increasing age and bachelor’s students, we also assumed, however, that lower stress levels would occur among people in relationships and living in the countryside. Materials and Methods: The study was conducted from 27 April 2022 to 12 May 2022. We chose that period as it was the final one of the COVID-19 pandemic, and there was an increasing amount of discussion concerning its cessation, with the public accustomed to its presence in our everyday lives. Since we wanted to determine the stress level experienced by students, we decided to use the standardized Perceived Stress Scale (PSS-10). We enriched the study with sociodemographic questions to investigate the potential impact of these characteristics on the degree of stress experienced. Results: The average score obtained by respondents on the PSS-10 was 19.57 ± 6.03. Of the respondents, 49% reported experiencing a high level of stress. No statistically significant differences were found between the mean PSS-10 scores and the gender (Z = 0.169; *p* = 0.865), age (F = 1.282, *p* = 0.281), marital status (Z = −0.776, *p* = 0.437), or place of residence (urban vs. rural) (Z = −0.784, *p* = 0.433) of the respondents. The mean PSS-10 scores were also analyzed regarding the level of education (bachelor’s vs. master’s). Bachelor’s students had an average PSS-10 score of 18.95 ± 6.42, while master’s students scored 20.05 ± 5.70. Again, no statistically significant differences were found (*t* = −1.102, *p* = 0.2720). Conclusions: The study indicated that nursing students experience high stress levels regardless of gender, age, marital status, place of residence, or level of education. High stress levels were reported among both bachelor’s and master’s students.

## 1. Introduction

In December 2019, a new threat emerged: the SARS-CoV-2 virus, first detected in Hubei Province, China. As disease and mortality rates rose in China and other countries, the World Health Organization’s Emergency Committee declared a global health emergency in January 2020. On 11 March 2020, the WHO declared COVID-19 a pandemic of unprecedented scale, affecting numerous countries worldwide. The pandemic has claimed over 7 million lives and resulted in more than 775 million recorded cases globally. The challenges posed by this new reality significantly impacted the healthcare system in Poland and worldwide, leading to stress and trauma among healthcare workers and medical students preparing to enter the profession [1,2,3,4,5].

The concept of stress originates from the Latin word “stringere”, meaning to squeeze, tighten, or pull tightly. Initially, it was primarily used in physics to describe the phenomenon of a centrally distributed force exerted on matter, causing stress. Over time, the term began to describe mental tension or negative factors leading to illness, with stress being seen as an external stimulus that conditions the occurrence of mental or physical dysfunction [6,7]. One of the first researchers to study the phenomenon of stress and its impact on human health was the Hungarian-Canadian endocrinologist Hans Selye. In 1936, Selye defined stress as a non-specific response of the body to various stressors, regardless of whether they are pleasant or unpleasant stimuli [8,9]. Stress can affect physical and mental health and human behavior. However, not every stressor is universal; what affects one person may be inconsequential to another. While we cannot eliminate stress, we can learn to avoid and manage it [8,9,10,11]. Stress can be understood in three dimensions: the first as a stimulus, in the form of everyday events causing tension and intense emotions; the second as a physiological or psychological reaction to a stressful situation; and the third as a process occurring between an individual and their environment [12].

The causes of stress can stem from various factors, known as stressors, such as the external environment, external stimuli, and events perceived by an individual as potential stress inducers [13]. These may include specific, challenging, and demanding situations that pose significant obstacles or threaten the safety of the individual and their loved ones [14]. Stressors can range from significant life events like marriage, divorce, relocation, starting school, pregnancy, childbirth, and the death of a loved one to everyday stressors like a broken-down car, being late for an appointment, or engaging in intensive physical effort [15,16]. While some of these events may have a positive aspect, they can still elicit fear and uncertainty, ultimately leading to stress [17].

Stress can have negative and positive effects on an individual, leading to a typology of stress types. The most commonly used classification, proposed by Hans Selye, distinguishes between distress and eustress. Distress is burdensome stress that overwhelms an individual’s coping mechanisms, potentially resulting in anxiety, withdrawal, and even depression. Eustress, conversely, is a positive form of stress that mobilizes an individual to act and enhances both mental and physical functioning. Selye also identified hyperstress, excessive stress, and hypostress, insufficient stress, noting that a balance between these is ideal, with eustress being dominant [8,10]. Excessive stress can lead to severe health issues such as strokes, heart attacks, stomach ulcers, and mental illnesses. In contrast, moderate stress can have positive effects, such as increased motivation, tolerance to the environment, and improvements in life achievements, such as sports performance [18,19,20].

Long-term stress has psychological effects and causes somatic symptoms in humans. These symptoms primarily result from the nervous system’s reactions to stressors and the subsequent biochemical processes, including the release of corticotropin, adrenaline, noradrenaline, and cortisol, commonly known as the stress hormone [21,22]. Symptoms associated with chronic stress may not always be immediately apparent or directly linked to stress. Still, they often recur in individuals exposed to stress, with a clear correlation: the higher the level of stress experienced, the greater the severity of symptoms.

The most common somatic symptoms resulting from prolonged exposure to stress include tension headaches, migraines, dizziness, tinnitus, chest pain, tachycardia, cardiac arrhythmias, hypertension, dyspnea, respiratory issues, abdominal pain, gastrointestinal disorders, diarrhea, bloating, constipation, nausea, changes in appetite, heartburn, urinary difficulties, allergies, eczema, hair loss, irregular menstruation, sleep disturbances, and chronic fatigue [23,24,25,26,27,28,29]. In addition to these somatic symptoms, individuals experiencing long-term stress may also exhibit self-destructive behaviors affecting their mental and social well-being. These behaviors include persistent anxiety, irritability, insomnia, avoidance of responsibility, difficulty forming relationships and partnerships, eating disorders, and substance abuse, such as smoking, alcoholism, and drug addiction [30,31].

Stress is directly linked to seven of the 10 leading causes of death worldwide, with occupational and organizational stress being significant risk factors for cardiovascular diseases [32]. Therefore, an important aspect of public health disease prevention is highlighting the destructive impact of stress on human health and life. How an individual responds to stress is influenced by numerous factors, including age, gender, intelligence level, personality traits, hereditary factors, cognitive predispositions, and life experiences [33]. These factors significantly affect the intensity of the stress response and complicate the diagnosis of stress, as there is no universal diagnostic tool applicable to every person and situation. The most commonly used questionnaires to assess stress levels include PSS-10, COPE, Mini-COPE, JSR, and DS14. These studies rely on questions where participants provide subjective responses based on specific point scales, which are then totaled to determine the stress level experienced [34,35].

An essential aspect of managing stress exposure is preventing and treating disorders resulting from it. While stress itself is not classified as a disease, it is a significant factor in developing stress-related diseases and disorders, such as post-traumatic stress disorder (PTSD). Given its impact on individuals, particularly concerning symptoms, diseases, and disorders, it is crucial to prioritize the early implementation of stress prevention strategies. Prevention can be approached at three levels:Primary prevention focuses on activities that enhance an individual’s ability to cope independently with stress, balance a private and professional life, and improve perception of the professional environment. Support from close individuals and society and positive everyday experiences play a vital role [36].Secondary prevention involves targeted interventions to assist individuals in self-regulating energy, emotions, and efficiency affected by stress [37].Tertiary prevention includes contact with counselors and therapeutic activities aimed at fully returning to functioning and health [38].

Given the destructive effects of long-term stress and the additional challenges presented by the COVID-19 pandemic, this study was undertaken. The pandemic posed new challenges to medical students, particularly nursing students, who observed nurses on the front lines facing unprecedented demands. These students, aware that the new reality also brought new challenges, were exposed to increased stress. Additionally, the restrictions introduced to limit the transmission of the SARS-CoV-2 virus, such as social distancing, isolation, and reduced social interactions, significantly impacted students’ stress levels. The shift to remote learning and reduced face-to-face contact often negatively affected the knowledge acquisition process, exacerbating stress and depressive moods due to the lack of direct human interaction.

Considering these experiences, we deemed it crucial to conduct this study to illustrate the extent to which the pandemic reality increased stress levels among nursing students. An important argument for conducting this study was that there were few studies available on this topic in Poland; we considered it important to include studies conducted in Poland in the global studies. Our study aimed to assess the impact of the COVID-19 pandemic during its last phase on the stress level of nursing students in Poland. We assumed that even though this was a period in Poland when public discussions increasingly suggested that the pandemic was coming to an end, with society beginning to accept the presence of coronavirus at a level similar to influenza, supported by protective vaccinations, stress levels would be high. Our objective was to determine the stress level experienced by students using a simple screening and standardized tool—the PSS-10 scale. Additionally, we aimed to analyze whether any sociodemographic factors, such as gender, age, place of residence, marital status, and level of academic education, were associated with increased stress levels among nursing students. 

For this purpose, we supplemented the study with sociodemographic questions. We chose to examine these aspects because we assumed that stress levels would be higher among women, older respondents, and bachelor’s students. We based our assumptions on the idea that women may be more sensitive and, therefore, may find it more challenging to cope with the pandemic, particularly in the context of social isolation and mental fatigue caused by the threat of infection and the potential loss of loved ones.

We also hypothesized that older respondents would experience greater stress compared to younger individuals. This assumption was grounded in the belief that younger people may have a lower sense of the value of life due to a constant search for extremes, less life experience, unstable life situations, and, often, the absence of offspring and family. Additionally, we assumed that bachelor’s students would experience higher levels of stress, as they tend to have less experience in dealing with patients, suffering, and death.

Conversely, we expected lower stress levels among people in relationships and those living in rural areas. Our rationale was that individuals in relationships can rely on greater emotional support from loved ones during difficult times, allowing them to share their problems, doubts, and experiences. Similarly, small communities, such as those in small towns and villages, are often well-integrated, with members supporting one another during hardships. In contrast, large cities are characterized by greater anonymity and indifference, which can contribute to feelings of alienation. This increasing anonymity can create a heightened sense of isolation, leading to additional stress as individuals feel they must face even the most challenging moments alone. This awareness can exacerbate tensions and reinforce feelings of hopelessness during difficult times.

Students who had already chosen a demanding professional path—one marked by a high risk of stress and trauma due to frequent exposure to human suffering and death—faced even greater demands on their physical and mental resilience during the global expansion of the SARS-CoV-2 virus. This study underscores the importance of providing adequate psychological care and support early on to protect the physical and mental well-being of individuals at heightened risk of stress. Such support is crucial for those who will form the future backbone of the healthcare system in Poland.

## 2. Materials and Methods

### 2.1. Participants and Study Characteristics

The study was conducted from 27 April 2022 to 12 May 2022, among 150 nursing students. The inclusion criterion was being a nursing student during the COVID-19 pandemic.

### 2.2. Method

The research tool used in the study was the standardized questionnaire, the Perceived Stress Scale (PSS-10), authored by Cohen S., Kamarck T., and Mermelstein R. The PSS-10 is designed to measure perceived stress levels and originally consisted of 14 questions, which were later reduced to 10 [39,40,41]. 

The study utilized the Polish adaptation of the PSS-10 scale by Juczyński Z. and Ogińska-Bulik N., published by the Psychological Testing Laboratory. In its Polish adaptation, the questionnaire consists of 10 questions assessing individual feelings related to events, problems, personal behaviors, and coping mechanisms [42]. Respondents answer the questions using a 5-point Likert scale (0-never, 1-almost never, 2-sometimes, 3-quite often, 4-very often). The maximum score in the Polish adaptation of the PSS-10 is 40 points; higher scores indicate a greater intensity of experienced stress. For an in-depth analysis, the obtained results should be converted to a sten scale and referred to an interval scale where, according to Polish standards developed for the PSS-10, scores of 1–4 sten indicate low stress, those of 5–6 sten indicate moderate stress, and those of 7–10 sten indicate high stress [34,43].

### 2.3. Statistical Analysis

The database and statistical analysis were performed using Statistica 9.1 software (StatSoft, Krakow, Poland). Quantitative variables were presented using the mean, standard deviation, and median, while qualitative variables were presented using the frequency and percentage.

To examine relationships between the analyzed variables for qualitative features, the Chi-squared (Chi^2^) test was used. The Shapiro–Wilk normality test assessed the normality of variable distributions in the analyzed groups.

Differences between the two groups were examined using the Student’s *t*-test; if the assumptions for this test were not met, the Mann–Whitney test was used. ANOVA (Analysis of Variance) with the Tukey RIR post hoc test was applied for comparisons involving three or more groups. If the assumptions for ANOVA were not met, the Kruskal–Wallis test was used.

A significance level of *p* < 0.05 was adopted to indicate statistically significant relationships or differences.

### 2.4. Ethical Statement

The study was conducted in accordance with the Declaration of Helsinki. All participants were informed about the purpose of the study and participated voluntarily and with informed consent.

## 3. Results

### 3.1. Sociodemographic Analysis of the Study Group

The characteristics of the study group are detailed in Table 1. The study population comprised 150 participants, primarily women (84.00%). The average age of the respondents was 37.6 years (SD = 10.4). The largest group in terms of residence consisted of individuals living in urban areas (57.33%). Most respondents were in a relationship (70.00%). Nearly all respondents (96.66%) were students at the State Academy of Applied Sciences in Chełm. The largest group by academic stage were second-cycle (master’s level) students (56%).

### 3.2. PSS-10 Questionnaire

Table 2 presents the results obtained by the examined in the Perceived Stress Scale (PSS-10); the data analysis showed that the average number of PSS-10 points obtained was 19.57 ± 6.03. In the study, no statistically significant differences were noted in the scope of sociogeographical characteristics such as gender, age, place of residence, marital status, or level of academic education between the compared groups and the obtained average PSS-10 result.

The results regarding the degree of stress experienced (Table 3) exemplified that 49.33% of the respondents experienced high-stress levels, with 12.00% reporting low levels. In terms of the quantitative interpretation of the results, which reflect the level of perceived stress in relation to sociodemographic characteristics, it was found that both men (45.83%) and women (50%) experience high stress levels. The analysis of respondents’ age in relation to perceived stress showed a high level of stress among those up to 30 years of age (58.70%) and those over 45 years of age (50.00%), while in the 31–45 age group, moderate stress levels are more common (46.77%). A high stress level was experienced by individuals living in urban areas (48.84%) and those in rural areas (50.00%). Similarly, in terms of marital status, both people in relationships (45.71%) and singles (57.78%) experience high levels of stress. Finally, both bachelor’s (45.45%) and master’s (52.38%) level students report similarly high levels of stress. Analysis of the stress levels concerning sociodemographic characteristics revealed no statistically significant relationships between the level of stress experienced and the analyzed variables.

The regression analysis (Table 4) did not show statistical significance; the analyzed predictors do not explain the obtained PSS-10 result.

## 4. Discussion

Our study aimed to assess the stress level experienced by nursing students in Poland during the final phase of the COVID-19 pandemic and to examine how the demographic characteristics of the participants moderated this stress level. At the outset, we hypothesized a correlation between students’ stress levels and factors such as gender, age, place of residence, marital status, and level of education. This study was necessary because there was insufficient literature analyzing this issue in Poland. We also anticipated that stress levels among students would be generally high, with higher levels occurring among women and bachelor’s students and increasing with age. In contrast, we expected lower stress levels among individuals in relationships and those living in rural areas. 

Nursing students, according to the body of evidence, are a group particularly vulnerable to experiencing increased stress [44]. It is mainly related to the feeling of pressure caused by the state of a public health emergency (Huang et al., 2020) [45], low knowledge of COVID-19 (Alsolais et al., 2021) [46], and remote learning (Masha’al et al., 2020 [47]; Matthes et al., 2022 [48]). The increased experience of stress by students is additionally intensified by the burden resulting from the large number of academic hours provided for medical fields, lack of time for rest and regeneration, extensive didactic material to master, and exhausting exams [48,49,50]. These factors contribute to the more frequent occurrence of symptoms in the form of higher rates of depression, insomnia, and suicidal thoughts [51,52]. Global studies conducted among nursing students have shown a tendency towards an increase in stress levels among this group as a result of the COVID-19 pandemic experiences (Gallego-Gómez et al., 2020 [53]; Hamadi et al., 2021 [54]; Mulyadi et al., 2021 [55]; Curcio et al., 2022 [56]). Our study confirms this trend by indicating that 49.33% of the students we examined presented a high stress level due to the COVID-19 pandemic, and only 12% experienced low-stress levels. The assumption we adopted and confirmed is confirmed by another Polish study (Bodys-Cupak et al., 2022 [57]), which showed a high level of stress (PSS-10) among 68.8% of Polish nursing students. Similarly, a study conducted among Iraqi nursing students, (Faraj, 2022 [58]) showed that 53.3% of the students presented an increased stress level (PSS-10) related to COVID-19. Also, a study of students in Saudi Arabia by Alayadi et al., 2024 [59] confirmed a high level of stress in 24.1% of the respondents. A different tendency to experience stress is manifested by respondents from China (Gao et al., 2020 [60]), where only 1.1% of the respondents showed increased stress during the COVID-19 pandemic.

The average result on the PSS-10 scale obtained by the respondents in our study is 19.57 ± 6.03. It confirms the tendency of the results obtained so far in global studies, including those in Turkey (Ersin i Kartal; 2021 [61], PSS-10: 17.61 ± 4.29) and New Zealand (Jagroop-Dearing; 2022 [62], PSS-10: 21.7 ± 7.1), and in a cross-national study by Ochnik et al., 2021 [63] that included Ukraine PSS-10: 19.93 ± 5.99, Czechia PSS-10: 18.16 ± 3.99, Slovenia PSS-10: 19.83 ± 7.56, and Russia PSS-10: 21.98 ± 6.95.

The analysis of the results of our research regarding the influence of gender on the level of perceived stress did not reveal any statistically significant differences in the obtained PSS-10 results (Z = 0.169 *p* = 0.865; Chi^2^ = 0.143, *p* = 0.930). The relationship between the influence of gender and the stress level that we assumed was not confirmed in our research. Despite the lack of confirmation of our hypothesis, the results we obtained are a valuable contribution to the current knowledge on this subject, confirming similar conclusions shown in other studies both Polish (Bodys-Cupak et al., 2022 [57]) and global (Faraj; 2022 [58] and Nebhinani et al., 2021 [64]). Results opposite to ours were presented in studies of students in Australia by Wynter et al., 2021 [65], Spain and South America by Espina-López et al., 2021 [66], Romania by Simionescu et al., 2021 [67], and Turkey by Ersin and Kartal; 2021 [61], which showed that there is a strong relationship between the occurrence of stress related to COVID-19 among women. A different result was shown in the study from China by Li et al., 2021 [68], which showed that men are more frequently exposed to more significant stress and the consequences of its experience on mental health. Our study does not sufficiently explain the reason for this state of affairs and the lack of confirmation of our hypothesis. Therefore, it would be worth repeating the study in the future on a larger, more diverse sample.

Similarly, our study did not demonstrate the assumed relationship between age and the level of perceived stress due to the COVID-19 pandemic. Our results confirm the absence of this relationship (H = 1.450, *p* = 0.484; Chi^2^ = 4.515, *p* = 0.340). This finding aligns with the global trend observed in studies that found no correlation between age and perceived stress levels. Similar conclusions were reported by Espina-López et al. (2021) [66], Bodys-Cupak et al. (2022) [57], and Iyigun et al. (2022) [69], who also found no relationship between age and the level of perceived stress due to COVID-19.

However, a different perspective was presented by Faraj (2022) [58] and Wynter et al. (2021) [65], who showed a correlation between greater stress levels and younger age. Additionally, Nebhinani et al. (2021) [64] suggested that stress levels related to COVID-19 were influenced by age. Opposite conclusions to our hypothesis and findings were reported by Gao et al. (2021) [60], who stated that older nursing students experienced lower stress levels than their younger counterparts.

The results we obtained in our study did not show the occurrence of the correlation we assumed between the level of stress experienced due to the COVID-19 pandemic and the marital status of the subjects (Z = −0.784, *p* = 0.432; Chi^2^ = 1.920, *p* = 0.382). Our research does not confirm the occurrence of the assumed statistical dependence in this respect; despite this, our study is a valuable contribution to the knowledge gathered on the impact of marital status on the stress level experienced. During the design and implementation of our study, we did not find any studies in the world literature on the phenomenon of stress in connection with the COVID-19 pandemic among nursing students that would analyze such a variable within their scope. Our study and the results obtained are new in this respect.

The hypothesis we adopted in our study was the influence of place of residence on perceived stress related to the COVID-19 pandemic. However, the results we obtained in the study did not confirm the occurrence of such a relationship (Z = −0.776, *p* = 0.437; Chi^2^ = 1.999, *p* = 0.368). The results we obtained contribute to the scope of knowledge on this subject, showing new trends and remaining in contrast to the studies by Gao et al., 2021 [60] and Simionescu et al., 2021 [67,70], which showed a relationship between the level of perceived stress related to the COVID-19 pandemic and the place of residence of the surveyed students. The cited studies also contradict each other in terms of the obtained results. The study by Gao et al., 2021 [60] showed a higher level of perceived stress among students living in rural areas, while Simionescu et al., 2021 [67,70] showed a greater tendency to experience stress among city dwellers. Our study would have contributed more to the knowledge gathered so far if it had been carried out on a more extensive and diverse sample. However, there is little research on the analyzed variable; therefore, our results are valuable.

The results we obtained regarding the occurrence of a correlation between the level of education at which students are and the level of perceived stress in connection with the experiences of the COVID-19 pandemic also did not confirm the relationship we assumed (t = −1.103, *p* = 0.271; Chi^2^ = 1.351, *p* = 0.509). Due to the deficit of knowledge and experience working with patients, we assumed that bachelor’s degree students would show a higher stress level than master’s degree students. The results we obtained confirm a similar tendency to that shown in other studies of Polish students (Bodys-Cupak et al., 2022 [57]), which showed that the level of perceived stress did not depend significantly on the year of study of the subjects. Our study is a valuable contribution to the existing knowledge, significant in that it is the second study carried out on a group of Polish students, which may confirm the occurrence of this type of phenomenon, i.e., the lack of influence of the level of education on the level of perceived stress. A different position to that of our research is presented in the studies of Wynter et al., 2021 [65], who show that a higher level of stress characterizes students who have completed more years of studies. A similar conclusion is stated in the studies of Aslan and Pekince; 2021 [71] and Ersin and Kartal; 2021 [61], who claim that the level of stress increased with a higher level of education among students; however, in both cases, no statistically significant relationship was demonstrated in this respect.

The study showed that the COVID-19 pandemic may be one of the stressors causing a deterioration in the mental condition of nursing students. 

The study revealed a high level of stress among the nursing students. However, it was not statistically associated with any of the analyzed sociodemographic variables, such as gender, age, place of residence, marital status, or stage of academic education. The lack of statistical significance does not imply that these variables do not influence the stress levels experienced by nursing students. As seen in studies conducted in various regions of the world, despite the absence of statistical significance, many findings are consistent with the hypotheses and results from similar research.

Our study demonstrates a notable trend in stress experienced by nursing students, which, as discussed earlier, aligns with findings from numerous studies conducted globally—this consistency is a strength of the study. Although we did not find statistical evidence linking the analyzed variables to stress levels in the context of our hypotheses, this does not detract from the value of the study.

### 4.1. Limitations

This study has its limitations. The limited sample size of 150 participants significantly affects the generalizability of the results. The duration of the study and the fact that it was conducted in a single academic institution also limited the number and diversity of participants. Five participants represented a different university, this being a study limitation in terms of representativeness. A multi-center study with a more extensive and diverse population of respondents would have provided more valuable insights into the research issue.

Additionally, the study could be expanded to include other important issues, such as the correlation between stress levels and academic performance, as well as the potential impact of stress on students’ educational outcomes. Another limitation is the use of the PSS-10 Scale, a screening tool designed to assess the degree of stress but does not evaluate the factors that predispose or mitigate its occurrence. Furthermore, the lack of comparative data prior to the COVID-19 pandemic regarding stress levels among the surveyed students could be perceived as another study limitation. Lastly, our study does not identify potential risk factors related to lifestyle, sleep patterns, diet, or physical activity that could correlate with stress levels, which is a significant limitation.

### 4.2. Implications

The high stress level among nursing students, as indicated by this study, underscores the need for further investigation into the underlying causes of this condition. Providing psychological and pedagogical support for students and assessing their coping strategies in high-stress situations is essential. Additionally, it is crucial to evaluate the impact of high stress on students’ mental health and the potential for complications such as anxiety disorders, depression, or post-traumatic stress disorder. 

A vital element of this process is to recognize the stress predictors enabling the development of prevention programs and their implementation within student education to counteract the consequences of stress. Those preventative measures are particularly crucial for medical professionals, who are often exposed to chronic trauma. It is essential to enable conditions within healthcare systems that provide access to psychological support and comprehensive care as part of mental health services. Those services should focus on early intervention, support, and diagnosis to address the consequences of experiencing elevated stress levels.

Future research should continue to explore the presented research problem while addressing these limitations. Additional standardized research tools could enhance the analysis of stress’s impact on mental health and coping mechanisms during high-stress situations.

Among the areas of future research, we should further explore the effects of the COVID-19 pandemic experiences on students’ mental health and its impact on the ability to cope with stress. In addition, it is vital to assess the impact of factors such as diet, physical activity, circadian rhythm, and lifestyle on the perceived level of stress in order to find potential causes that enhance and alleviate the level of perceived stress. These aspects will help build strategies for mental health programs when experiencing stress.

Future research should focus on these areas, emphasizing the importance of timely therapeutic interventions to mitigate the effects of stress on human health.

## 5. Conclusions

The study revealed that, among nursing students in Poland, the COVID-19 pandemic was a significant, unprecedented, and unpredictable stressor that led to a substantial increase in stress levels. However, our results did not demonstrate a statistical relationship between the analyzed sociodemographic variables and stress levels. Specifically, the study did not confirm the hypotheses that women and bachelor’s students would experience higher stress levels, nor did it confirm that stress would increase with age. Factors we initially believed would contribute to higher stress levels were found to be neutral in this context.

Additionally, our assumptions regarding the protective effect of certain factors in reducing stress were not confirmed. Respondents in relationships and those living in rural areas did not show the expected buffering effect against stress experienced during the pandemic. This study offers a new perspective on the issue of stress among nursing students in Poland. This topic has largely been marginalized and underestimated, especially in terms of its mental health consequences.

The COVID-19 pandemic has exposed new threats, such as fear of infection and death, work overload, social isolation, and the unique challenges posed by the nursing profession. Our research highlights how these factors affected the ability to cope with stress and the degree to which it was perceived. Despite finding high stress levels, we observed no significant influence of the analyzed variables on these stress levels. This underscores the need for ongoing monitoring of stress among nursing students and for early interventions to protect their mental health.

Our study, being one of the few conducted in Poland during the COVID-19 pandemic, emphasizes how much this issue has been overlooked, with the impact of stress on nursing students being greatly underestimated. Therefore, it may be worth conducting further research in the near future, with a larger and more selective sample, to deepen the analysis of stress among nursing students in Poland.

## Figures and Tables

**Table 1 healthcare-12-01885-t001:** Sociodemographic analysis of the study group.

Variable	Category	Number (N)	Percentage (%)
Gender	Woman	126	84.00
	Man	24	16.00
Age	Up to 30 years of age	46	30.67
	31–45 years of age	62	41.33
	Over 45 years of age	42	28.00
Place of residence	Urban area	86	57.33
	Rural area	64	42.67
Marital status	In relation with	105	70.00
	Single	45	30.00
Place of study	State Academy of Applied Sciences in Chełm, Poland	145	96.67
	University of Warmia and Mazury in Olsztyn, Poland	1	0.67
	Medical University of Lublin, Poland	4	2.67
Level of academic education	Bachelor	66	44.00
	Master	84	56.00

**Table 2 healthcare-12-01885-t002:** Differences in PSS-10 score in relation to the sociodemographic characteristics of the subjects.

Variable	Category	M	Me	SD	Statistical Analysis
Gender	Woman	19.44	19.50	5.83	Z = 0.169 *p* = 0.865
	Man	20.21	19.00	7.07	
Age	Up to 30 years of age	20.35	20.00	5.78	H = 1.450 *p* = 0.484
	31–45 years of age	19.64	19.00	6.28	
	Over 45 years of age	18.59	19.50	5.92	
Place of residence	Urban area	19.30	19.00	5.90	Z = −0.776 *p* = 0.437
	Rural area	20.18	20.00	6.35	
Marital status	In relation with	18.99	19.00	6.27	Z = −0.784 *p* = 0.432
	Single	20.34	19.50	5.64	
Level of academic education	Bachelor	18.95	19.00	6.42	*t* = −1.103 *p* = 0.271
	Master	20.05	20.00	5.670	

M—average, Me—median, SD—standard deviation, H—Kruskal–Wallis test, Z—Mann–Whitney test, *t*—Student’s *t*-test, *p*—statistical significance.

**Table 3 healthcare-12-01885-t003:** Relationship between the level of stress experienced and sociodemographic characteristics of respondents.

Variable	Category	The Level of Stress Experienced						Chi^2^ *p*
		Low 1–4 Sten Score		Average 5–6 Sten Score		High 7–10 Sten Score		
		n	%	n	%	n	%	
Gender	Woman	15	11.90	48	38.10	63	50.00	Chi^2^ = 0.143 *p* = 0.930
	Man	3	12.50	10	41.67	11	45.83	
Age	Up to 30 years of age	4	8.70	15	32.61	27	58.70	Chi^2^ = 4.515 *p* = 0.340
	31–45 years of age	7	11.29	29	46.77	26	41.29	
	Over 45 years of age	7	16.67	14	33.33	21	50.00	
Place of residence	Urban area	13	15.12	31	36.05	42	48.84	Chi^2^ = 1.999 *p* = 0.368
	Rural area	5	7.81	27	42.19	32	50.00	
Marital status	In relation with	13	12.28	44	41.90	48	45.71	Chi^2^ = 1.920 *p* = 0.382
	Single	5	11.11	14	31.11	26	57.78	
Level of academic education	Bachelor	10	15.15	26	39.39	30	45.45	Chi^2^ = 1.351 *p* = 0.509
	Master	8	9.52	32	38.10	44	52.38	
Total		18	12.00	58	38.67	74	49.33	-

**Table 4 healthcare-12-01885-t004:** Regression analysis of PSS-10 score and sociodemographic characteristics.

Variable	PSS-10 R^2^ Adjusted = 0.006 F = 0.856 *p* = 0.529			
	b Factor	Standardized b	*t*	*p*
The word Free	16.526		4.580	<0.001
Gender	−0.034	−0.059	−0.664	0.508
Age	1.241	0.076	0.869	0.386
Place of residence	1.565	0.129	1.512	0.133
Marital status	−0.993	−0.076	−0.865	0.389
Level of academic education	2.265	0.187	0.913	0.363

## Data Availability

The data presented in this study are available to all researchers.

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
