# Peer review of "Stress among Nursing Students in the Era of the COVID-19 Pandemic"

_healthcare, 2024, doi:10.3390/healthcare12181885_

Round 1

Reviewer 1 Report

Comments and Suggestions for Authors

The article provides a comprehensive overview of the impact of the COVID-19 pandemic on stress levels among nursing students in Poland. It effectively explores the concept of stress, its causes, effects, and management strategies. The research methodology is well-explained, and the findings are presented in a clear and concise manner.

The article provides a detailed review of existing literature on stress, its causes, and effects, particularly in the context of healthcare professionals. The research methodology is clearly outlined, including the sample size, data collection methods, and statistical analysis techniques. The findings are directly related to the research question and provide valuable insights into the stress experienced by nursing students during the pandemic. The article is well-written and easy to understand, making it accessible to a wide audience.

In the methodology, in addition to the quantitative PSS-10, including open-ended questions in the survey to gather qualitative data on the specific stressors experienced by the students and their coping mechanisms. Also, conduct follow-up assessments to track changes in stress levels over time and evaluate the effectiveness of any interventions implemented. Examine the correlation between stress levels and academic performance to assess the impact of stress on students' educational outcomes. Investigate the relationship between stress levels and health behaviors, such as sleep patterns, diet, and exercise, to identify potential risk factors.

Consider these additional points to enhance the analysis and results section of the article:

·       Divide the sample into subgroups based on sociodemographic factors (e.g., age, gender, place of residence) and analyze differences in stress levels within these groups.

·       Explore if the impact of the pandemic on stress levels varies based on the combination of different sociodemographic factors (e.g., age and gender).

·       Analyze the qualitative data using thematic analysis to identify common themes and patterns related to stress.

·       Use regression analysis to examine the relationship between stress levels and various predictor variables.

The discussion session can be improved by:

·       Exploring whether certain factors (e.g., coping strategies, social support) mediate or moderate the relationship between the pandemic and stress levels.

·       If possible, comparing the stress levels of the current nursing students with data from the same population before the pandemic. This would provide a stronger baseline for assessing the impact of COVID-19.

·       Limitations: The article could benefit from a more explicit discussion of the study's limitations, such as the sample size and potential for selection bias.

·       Discussion of Implications: The discussion section could delve deeper into the implications of the findings for nursing education, healthcare systems, and mental health services.

·       Future Research Directions: The article could suggest potential areas for future research to further explore the impact of the pandemic on nursing students and to develop effective interventions.

Overall, this article is a valuable contribution to the literature on the impact of the COVID-19 pandemic on healthcare professionals. It provides important insights into the stress experienced by nursing students and highlights the need for interventions to support their mental health and well-being.

Reviewer 2 Report

Comments and Suggestions for Authors

C1: Thanks for allowing me to review the manuscript entitled "Stress among Students Nursing in the era of the COVID-19 pandemic". While I find the study interesting, I noted some elements that are in need of improvement. PLease, refer to the following comments.

C2: PLease avoid citations in the abstract. Please include why you decided to control for the effect of individual differences and the reason for doing the data collection April 27, 2022, to May 12, 2022.

C3: The introduction presents a series of elements that appear to be interesting for prevention programs, and this is an apsect that I really value. However, I must recognize that the introduction does not present the point of the article, what is the purpose? What are the reasons for studying the role of individual differences? PLease, provide substantial arguments to sustain your methodology and analysis.

C4: Data and results are well presented. All good!

C5: The discussion does not take into account the study but takes a lot of space to present the impact of Covid, and stress. Please, rewrite the discussion in order to present a) the aim of the study, b) its rationale, c) the main results. Then, discuss how your results provide theoretical contributions to the existing literature. 

C6: You may consider splitting the discussion into a) theoretical contributions, b) applied implications and c) limitations.

C7: The conclusion repeats what has already been said. Please explain the novelty of your study.

Comments on the Quality of English Language

I think the English is good, but sentences could be improve since the wording appear to be not very clear sometimes.

Round 2

Reviewer 2 Report

Comments and Suggestions for Authors

Thanks for (your limited) working on the manuscript. In contrast to your response letter, I noted that you did not address the points (e.g., putting references in the abstract) but were limited to rephrasing and including some elements. 

This is not a joke.

Please revise your paper following my previous comments. In particular, I would like to encourage you to address the following main problems:

1) The introduction presents a series of elements that appear to be interesting for prevention programs, and this is an apsect that I really value. However, I must recognize that the introduction does not present the point of the article, what is the purpose? What are the reasons for studying the role of individual diFerences? PLease, provide substantial arguments to sustain your methodology and analysis

2) The discussion does not take into account the study but takes a lot of space to present the impact of Covid, and stress. Please, rewrite the discussion in order to present a) the aim of the study, b) its rationale, c) the main results. Then, discuss how your results provide theoretical contributions to the existing literature

If you do not find my comments helpful, please explain to me and the editors why. 

I am waiting for your revised version.

Best

Comments on the Quality of English Language

Minor issues
